# Contact-number-driven virus evolution: A multi-level modeling framework for the evolution of acute or persistent RNA virus infection

Junya Sunagawa[1☯], Ryo Komorizono[2☯], Hyeongki Park[3☯], William S. Hart[4], Robin N. Thompson[5,6], Akiko Makino[2,7], Keizo Tomonaga[2,7,8‡], Shingo Iwami[3,9,10,11,12,13‡*], Ryo Yamaguchi[1,14*]

**1** Department of Advanced Transdisciplinary Science, Hokkaido University, Sapporo, Hokkaido, Japan, **2** Laboratory of RNA Viruses, Department of Virus Research, Institute for Life and Medical Sciences (LiMe), Kyoto University, Kyoto, Japan, **3** interdisciplinary Biology Laboratory (iBLab), Division of Natural Science, Graduate School of Science, Nagoya University, Nagoya, Japan, **4** Mathematical Institute, University of Oxford, Oxford, United Kingdom, **5** Mathematics Institute, University of Warwick, Coventry, United Kingdom, **6** Zeeman Institute for Systems Biology and Infectious Disease Epidemiology Research, University of Warwick, Coventry, United Kingdom, **7** Laboratory of RNA Viruses, Graduate School of Biostudies, Kyoto University, Kyoto, Japan, **8** Department of Molecular Virology, Graduate School of Medicine, Kyoto University, Kyoto, Japan, **9** Institute of Mathematics for Industry, Kyushu University, Fukuoka, Japan, **10** Institute for the Advanced Study of Human Biology (ASHBi), Kyoto University, Kyoto, Japan, **11** Interdisciplinary Theoretical and Mathematical Sciences Program (iTHEMS), RIKEN, Saitama, Japan, **12** NEXT-Ganken Program, Japanese Foundation for Cancer Research (JFCR), Tokyo, Japan, **13** Science Groove Inc., Fukuoka, Japan, **14** Department of Zoology & Biodiversity Research Centre, University of British Columbia, Vancouver, British Columbia, Canada

☯ These authors contributed equally to this work.
‡ KT and SI also contributed equally to this work.
* iwami.iblab@bio.nagoya-u.ac.jp (SI); ryamaguchi@sci.hokudai.ac.jp (RY)

## Abstract

Viruses evolve in infected host populations, and host population dynamics affect viral evolution. RNA viruses with a short duration of infection and a high peak viral load, such as SARS-CoV-2, are maintained in human populations. By contrast, RNA viruses characterized by a long infection duration and a low peak viral load (e.g., borna disease virus) can be maintained in nonhuman populations, and the process of the evolution of persistent viruses has rarely been explored. Here, using a multi-level modeling approach including both individual-level virus infection dynamics and population-scale transmission, we consider virus evolution based on the host environment, specifically, the effect of the contact history of infected hosts. We found that, with a highly dense contact history, viruses with a high virus production rate but low accuracy are likely to be optimal, resulting in a short infectious period with a high peak viral load. In contrast, with a low-density contact history, viral evolution is toward low virus production but high accuracy, resulting in long infection durations with low peak viral load. Our study sheds light on the origin of persistent viruses and why acute viral infections but not persistent virus infection tends to prevail in human society.

**Data Availability Statement:** All codes necessary to repeat the simulation described in this study have been made available. Python source codes

are hosted on Github (https://github.com/j-s9vcp/ContactNumberDriven_VirusEvolution) [69]. There are no data to be archived.

**Funding:** This study was supported in part by Grants-in-Aid for JSPS Scientific Research (KAKENHI) Scientific Research B 18KT0018 (to S. I.), 18H01139 (to S.I.), 16H04845 (to S.I.), Scientific Research in Innovative Areas 20H05042 (to S.I.), 19H04839 (to S.I.), 18H05103 (to S.I.), 21K15160 (to R.Y.), JP20H05682 (to K.T.), JP21K19909 (to K.T.); JSPS Overseas Research Fellowships (to R.Y.); ACT-X JPMJAX22AK (to R. Y.); AMED CREST 19gm1310002 (to S.I.); AMED Research Program on HIV/AIDS 19fk0410023s0101 (to S.I.); AMED Japan Program for Infectious Diseases Research and Infrastructure, 20wm0325007h0001, 20wm0325004s0201, 20wm0325012s0301, 20wm0325015s0301 (to S.I.); AMED Research Program on Emerging and Re-emerging Infectious Diseases 19fk0108156h0001, 20fk0108140s0801 and 20fk0108413s0301 (to S.I.); AMED Program for Basic and Clinical Research on Hepatitis 19fk0210036h0502 (to S.I.); AMED Program on the Innovative Development and the Application of New Drugs for Hepatitis B 19fk0310114h0103 (to S.I.); Moonshot R&D Grant Number JPMJMS2021 (to S.I.) and JPMJMS2025 (to S.I.); JST MIRAI (to S.I.); Mitsui Life Social Welfare Foundation (to S.I.); Shin-Nihon of Advanced Medical Research (to S. I.); Suzuken Memorial Foundation (to S.I.); Life Science Foundation of Japan (to S.I.); SECOM Science and Technology Foundation (to S.I.); The Japan Prize Foundation (to S.I.), and Daiwa Securities Health Foundation (to S.I.). The funders had no role in study design, data collection and analysis, decision to publish, or preparation of the manuscript.

**Competing interests:** The authors have declared that no competing interests exist.

## Author summary

As exemplified by the SARS-CoV-2 variants of concern and influenza A virus variants, we need to predict the future evolution of viral properties to counter an oncoming pandemic. While some RNA viruses such as influenza A viruses adopt an acute infection strategy, others, such as Borna disease virus, adopt a persistent infection strategy. In what kind of environment did these viruses evolve? This study presents a modeling framework to investigate the evolution of proliferative ability and accuracy throughout the viral life cycle. Specifically, we calculated eco-evolutionary dynamics from virus infection in the host to transmission between hosts and capture the transmission potential of the virus. The acute infection phenotype evolves under a contact history involving frequent contacts between hosts. By contrast, the persistent infection phenotype evolves under an environment with a small mean and a large variance in the host contact history. These findings have direct implications for the fight against the continuous evolution of various viruses, which have easily prevailed in the setting of dense human contacts. We believe it is essential to readdress the evolution of viruses from an evolutionary and ecological perspective.

## Introduction

Because they lack proofreading activity in their RNA-dependent RNA polymerase, RNA viruses exhibit extremely high mutation rates, generating $10^{-6}$ to $10^{-4}$ substitutions per nucleotide, orders of magnitude greater than those of most DNA-based life forms [1–3]. The accuracy (or inaccuracy) of the polymerase is the source of intra-host viral diversity and determines viral propagation and virulence *in vivo* [4,5].

In general, acute infectious RNA viruses, such as SARS-CoV-2, are highly proliferative and release large numbers of progeny viral particles [6]. From an epidemiologic point of view, acute infectious RNA viruses are highly pathogenic [7,8]. Infections with most non-retroviral RNA viruses result in the characteristic symptoms and signs of acute illness. On the other hand, persistent viral infection is a different strategy. The properties of RNA viruses vary widely, among which non-retroviral RNA viruses can often be broadly classified according to acute infection and persistent (chronic) infection. Some non-retroviral RNA viruses, such as hepatitis C virus, coxsackievirus, and borna disease virus, establish persistent infection and replicate in the host for long periods [9]. Whereas acute infectious viruses are eliminated by the immune response within a few weeks, persistent infectious viruses are maintained in the host for at least several months or even years. In most cases, patients are mildly symptomatic or asymptomatic during persistent infection. Given the multiple viral and host factors involved in viral pathogenesis, it is not easy to discern the causes of mortality in the course of a viral infection. Many acutely infectious RNA viruses cause fatal disease soon after infection (e.g., rabies virus, Nipah virus), whereas some persistent infectious RNA viruses cause nonfatal disease after infection (e.g., Borna disease virus, Pegi virus) [10]. A persistent infectious virus can coexist with the host by reducing the amount of progeny virus released from the infected cells, slowing down the replication rate, decreasing the evolutionary rate, or evading the host's immune system to use the host as its own reservoir until the next opportunity to infect [11–15]. These different viral phenotypes of acute and persistent infection express different strategies in the host population, ranging from intracellular replication rates and innate immune response to inter-individual transmission dynamics [9].

For gaining a better understanding of how viral properties are optimized, the lifestyles of the host organisms are of critical importance. Different host behaviors provide different environments, with some environments favoring particular viral characteristics. Human activities in the Anthropocene have greatly affected the global environment, increasing the frequency of human-animal contact and various human-induced evolutionary changes in animals [16]. Traditionally, it is considered that acute infection evolved because of population-level transmission potential, that is, the potential for many susceptible hosts to come into contact with an infected individual [17]. This suggests the question, "Why did persistent infectious RNA viruses evolve?" Commonly, the population-level transmission potential is characterized by the (basic) reproduction number [8]. However, the environmental conditions in the aspect of contact history that favor the evolution of persistent infectious RNA viruses have not yet been explored. In addition, in terms of the within-host virus life cycle, it is unclear how viral properties such as the proliferation rate and the accuracy of viral replication affect virus evolution under different environmental conditions. Another unresolved issue is why acute infections are generally more prevalent than persistent infections in the current profile of viral diversity in human populations [11].

To tackle these challenges, we employed a multi-level modeling approach including both individual-level virus infection dynamics and population-scale transmission. Because we aimed to seek an optimal strategy across a broad range of viruses rather than explain the properties of a specific virus, we simply assume that an optimization occurs at an epidemiologic equilibrium (i.e., we focus on equilibrium solutions). Although elaborate discussions are made possible by considering the (basic) reproduction number and balance with other high-level population criteria like lifespan of the host in models specific to each virus (e.g., [18,19]; also see [20] as an example for limitations to the idea that natural selection will always maximize the number), we examine the long-term equilibrium evolutionary outcome in the context of stable endemic diseases which solely maximizes "transmission potential" as the reproduction number (i.e., the total number of secondary cases generated throughout the infectious period). Our simplification is valid unless considering the situation where pathogens often showing non-equilibrium dynamics such as antigenic escape from host immune defenses through evolution [21]. Under different environments characterized by different contact numbers in populations, we evaluated how measures of the proliferation rate and accuracy in the viral life cycle affect population-level transmission potential and therefore RNA virus evolution.

## Results

### Virus evolution in different contact scenarios

To explore how virus evolution is influenced by the environment, we first developed a probabilistic multi-level model to characterize population-level transmission potential, $R_{TP}$. Using a genetic algorithm (see **Materials and Methods** for further details), we investigated the evolution of the proliferation rate and the accuracy of replication in the viral life cycle, $p$ and $\varepsilon$, respectively. Our model [Eqs (7–8)] calculates $R_{TP}$ as the sum of secondary cases generated throughout the infectious period by a primary case (see **Fig 1A**, **Materials and Methods** in detail). Calculating $R_{TP}$ means that we investigate the virus evolution as a long-term equilibrium (i.e., population-level transmission potential) here rather to explore non-equilibrium co-evolutionary dynamics with host individuals. The duration of the infectious periods $D(V_{\text{total}})$ is assumed to depend on the total viral load. Specifically, the duration of the infectious period decreases with the cumulative viral load. In each parameter combination (i.e., $p$ and $\varepsilon$), $D(V_{\text{total}})$ is calculated by Eqs (5–6) based on the virus infection model [Eqs (1–4)] with other fixed

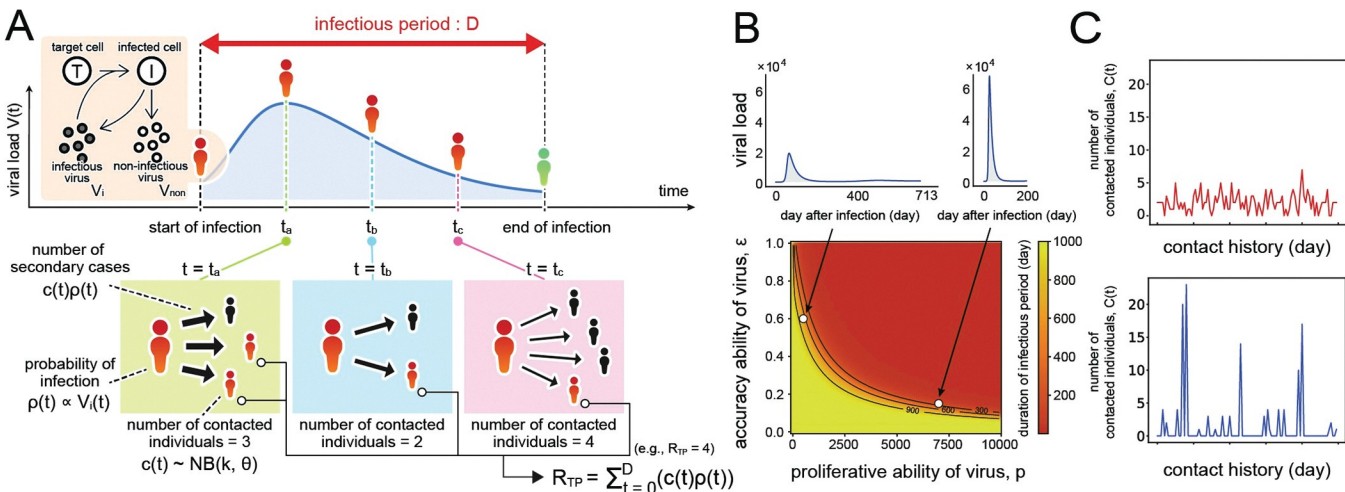

**Fig 1. Schematic illustration of multi-level disease transmission.** A schematic of our model is depicted in (**A**). The probability of infection $\rho(t)$ of susceptible individuals from an infected hosts depends on their viral load $V_i(t)$. At each time step (i.e., day), the focal infected individual has contacts with multiple susceptible individuals. Here the contact numbers $C(t)$ are assumed to follow a negative binomial distribution, which does not depend on the viral load. The sum of newly infected individuals (i.e., [number of contacted individuals per day] × [probability of infection per contacted individual]) during the infectious period is calculated as $R_{TP}$. The duration of the infectious periods for each parameter combination ($p$ and $\varepsilon$) characterizing the proliferative ability and accuracy of viruses are calculated in (**B**) based on the virus infection model. The left panel corresponds to an infectious period of 713 days with a relatively low viral load, and the right panel represents an infectious period of 200 days with a high viral load. Examples of generated daily contact numbers characterizing environmental conditions are shown in (**C**). The contact numbers under a negative binomial distribution with $k =100$ and $\theta =0.03$ (the mean and variance are 3.0 and 3.1) and $k = 0.12$ and $\theta = 10$ (the mean and variance are 1.2 and 13.3) are shown in the top and bottom panels, respectively.

parameters (see **Table A in S1 Text**). The calculated infectious period for each parameter pair is described in **Fig 1B**.

   To evaluate how the proliferative ability and accuracy of replication of viruses are optimized to increase the transmission potential, we assumed that environments are characterized by the number of contacts per day. We made two categories of contact history with daily numbers of contacts sampled from the negative binomial distribution (see **Materials and Methods** in detail). The first category is one with large mean and small variance, in which the contact number is always around the mean throughout the contact history and does not decrease too low or increase too high (**Fig 1C**, top). The other has small mean and large variance, in which the contact number is usually 0 but is sometimes very high (**Fig 1C**, bottom). For contact numbers with large mean and small variance, $k = 100$ and $\theta = 0.03$ (the mean number is 3) are fixed to generate the contact history. In the history with small mean and large variance, $k = 0.12$ and $\theta = 10$ (the mean number is 1.2). Here $k$ and $\theta$ represent the shape parameter and scale parameter of the negative binomial distribution, respectively.

   Given a contact history generated from the negative binomial distribution, as described above, we searched for the optimal point in the fitness landscape (i.e., the largest $R_{TP}$), that is, the optimal combination of $p$ and $\varepsilon$, by using a genetic algorithm for each contact number (see **Materials and Methods** in detail). For the contact history with large mean and small variance (i.e., high-density contact history: ($k$, $\theta$) =(100, 0.03), **Fig 1C**, top), optimal fitness involves high virus production but low accuracy, which results in a short duration of infectious period with high peak viral load (we call this the *acute infection* phenotype) (**Figs 2A** and **2D in S1 Text**). In contrast, for the contact history with small mean and large variance (i.e., low-density contact history: ($k$, $\theta$) = (0.12, 10), **Fig 1C**, bottom), the virus is more likely to be optimized toward low virus production but high accuracy, resulting in a long infection duration with low peak viral load (we call this the *persistent infection* phenotype) (**Figs 2C** and **D in S1 Text**).

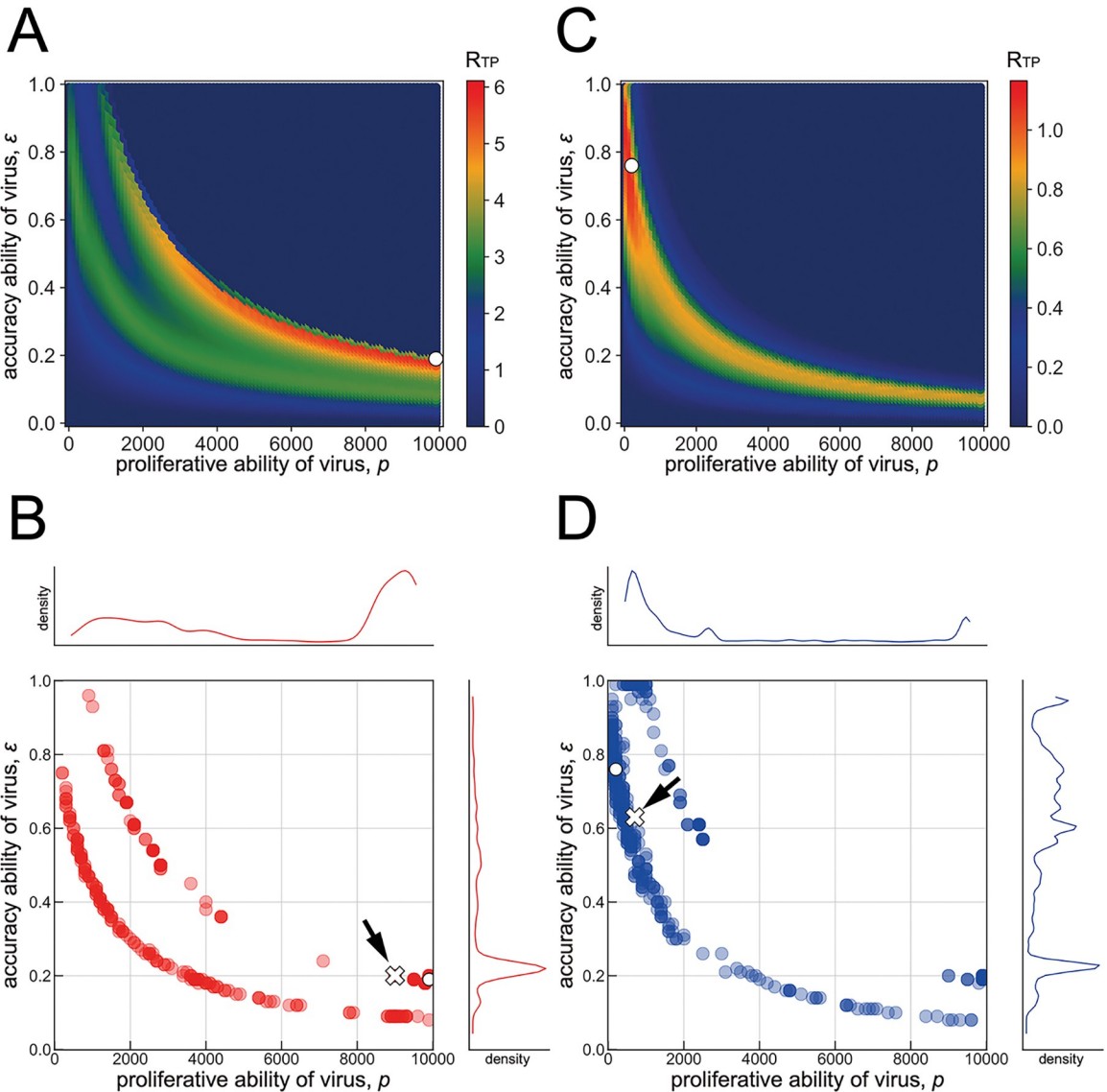

**Fig 2. Virus evolution in a given environment.** The optimal parameter combinations ($p$ and $\varepsilon$) that maximize $R_{TP}$ are calculated for a parameter set of $k = 100$ and $\theta = 0.03$ in (**A**). The white dot represents the optimal point of $R_{TP}$ in a single simulation. Using the same parameter set, 500 optima based on independent contact histories are calculated in (**B**). The kernel plots illustrated alongside the top and right parts are the marginal distributions of $p$ and $\varepsilon$, respectively. The white dot exactly corresponds to the optimal point shown in a single simulation in (**A**). The white cross indicated by the black arrow corresponds to the median of optimized parameter set of $p$ and $\varepsilon$ based on the 500 simulation runs. In a similar manner, the parameter set of $k = 0.12$ and $\theta = 10$ is used to calculate the optimal parameter combinations ($p$ and $\varepsilon$) in (**C**). Using the same parameter sets, 500 evolutionary consequences for independently drawn contact histories are shown in (**D**). The white dot and white cross correspond to the optimal point of the parameter set and the median point, respectively.

The white dots in **Fig 2A and 2C** correspond to the coordinates of the combination of $p$ and $\varepsilon$ that maximize $R_0$ with a given contact pattern.

With 500 independent iterations of this process with the same parameter sets, the evolutionary endpoint of each iteration is distributed as shown in **Fig 2B and 2D**. The white dots in **Fig 2B and 2D** correspond to the evolutionary endpoint in each single simulation run, **Fig 2A** and **Fig 2C**, respectively. The white crosses indicated by the black arrows in **Fig 2B and 2D**

correspond to the median parameter set of $p$ and $\varepsilon$ after evolution under the given environment sequence of the contact history.

In addition to these contact histories, we considered the evolution of viral phenotype under other contact histories in scale-free networks and small-world networks. In the analysis on networks, we interpret the evolutionary consequences of the phenotypes similarly. Under networks where hubs exist or where individuals are in contact with each other evenly and with high frequency, the average value of the contact history is stable at a high value, and the acute infection type is likely to be optimal, and vice versa (**Figs E** and **F in S1 Text**). As described in the next section, we investigated how the trend of virus evolution changes under the different environments.

## Virus evolution depending on the difference in mean contact number

We explored the optimal properties of the virus (proliferation rate and accuracy of replication, $p$ and $\varepsilon$, respectively) for transmission by gradually changing the mean number of daily contacts in the contact history. To represent various contact histories, we explored a total of 441 patterns of contact histories ranging from 0.01 to 100. That is, we used the following parameter sets: shape parameter $k$ ranges from $10^{-1.0}, 10^{-0.9}, 10^{-0.8}, \ldots, 10^{0.9}, 10^{1.0}$ (sum up to 21 points), as is also the case with scale parameter $\theta$, and each contact history is made by the combination of each $k$ and $\theta$. We conducted a total of 500 independent iterations for each parameter set of $(k, \theta)$ and determined the evolutionary consequences of the median parameter set of $p$ and $\varepsilon$.

We found that viruses are optimized toward low production but high accuracy (the persistent infection phenotype) in regions of low-density contact history (the mean contact number from 0.01 to 1) regardless of the variance in daily contacts (white dots 3 and 4 in **Fig 3A and 3B**). As discussed above, these optimized virus infection dynamics showed a longer duration of infectious period with low peak viral load (white dots 3 and 4 in **Fig 3C and 3D**). In contrast, in the region of a highly dense contact history ranging from 30 to 100, viruses were optimized with high production but low accuracy (white dots 1 and 2 in **Fig 3A and 3B**), representing the acute infection phenotype with the shorter duration of infectious period and high peak viral load (white dots 1 and 2 in **Fig 3C and 3D**). In addition, it is noteworthy that large variance in daily contacts can select for greater accuracy/lower proliferative capacity (**Fig 4A and 4B**). Viruses that adapted to the highly variable environment were characterized by a persistent infectious phenotype and a low transmission potential, especially under a small mean contact number (**Fig 4C and 4D**). With respect to the transmission potential, $R_{TP}$, a logarithmic increase is observed as the mean contact number increases (**Fig 3E**). Note that some of the persistent infection phenotypes evolving under the low-density contact history (white dot 4 in **Fig 3A, 3B, 3D and 3E**) might have a low transmission potential characterized by $R_{TP} < 1$ (see **Discussion**).

Importantly, we demonstrated that, if the contact history is of intermediate density (with mean contact numbers ranging from 1 to 30), viruses can be optimized toward either acute or persistent infection, showing a critical transition on the virus infection phenotypes (**Fig 3D**). This bistable-like pattern may appear by chance owing to the stochasticity of contact number history, even when drawn from the same distribution. Further discussion is provided below.

## Discussion

In this study, we investigated the impact of the proliferation rate and the accuracy of replication on viral evolution by considering optimal combinations for transmission under different contact histories of a host organism. Specifically, assuming an epidemiologic equilibrium, we developed a probabilistic multi-level model that characterizes population-level transmission

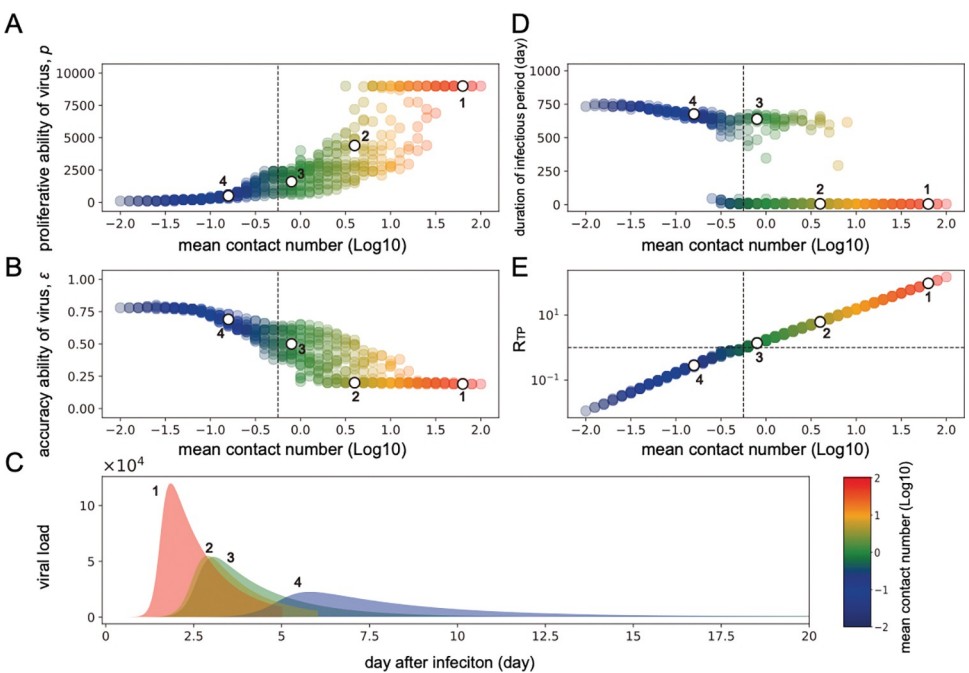

**Fig 3. Virus evolution in various scenarios of contact history.** A total of 441 contact history patterns are explored to determine the viral evolution to an optimal $R_{TP}$ in a given environmental scenario. 500 iterations of each parameter combination of $k$ and $\theta$ are conducted to determine the optimized parameter set of $p$ and $\varepsilon$ characterizing final virus infection dynamics (i.e., different viral infection phenotypes). The median of the optimized parameter set of $p$ and $\varepsilon$ based on the 500 simulation runs is shown against the mean contact number of histories (with logarithmic increase) determined by each parameter set of $k$ and $\theta$. The combination of $p$ and $\varepsilon$ is plotted in **(A)** and **(B)**, separately. Colors represent the degree of the mean contact number: blue corresponds to a low mean contact number, whereas red corresponds to a high mean contact number. How virus infection dynamics vary with the mean contact number is shown in **(C)**. Several examples of the virus infection dynamics are shown to represent the difference in the infectious period and the peak viral load. Numbers 1, 2, 3, and 4 in **(C)** are the corresponding dynamics illustrated as white dots in **(A)**, **(B)**, **(D)**, and **(E)**, respectively. The duration of the infectious period and $R_{TP}$ varied with the mean contact number are plotted in **(D)** and **(E)**, respectively. In **(E)**, the vertical dashed line separates the area of mean contact number by an index, $R_{TP} = 10°(= 1)$. The corresponding dashed line is also illustrated in **(A)**, **(B)**, and **(D)**.

potential based on individual-level virus infection dynamics. The result presented above show that, depending on host density, acutely infectious viruses acquire a phenotype of a high mutation rate and a high replication rate, whereas persistently infectious viruses acquire a phenotype of a low mutation rate and a slow replication rate. A similar phenomenon has been observed in relation to the characteristics of actual RNA viruses. Borna disease virus (BDV), a persistent infectious RNA virus, has been reported to have a suppressed intracellular replication speed and a very slow evolutionary rate [22,23]. The number of virus particles released from BDV-infected cells is also extremely low [24]. Moreover, when coronaviruses shift their phenotype to persistent infectivity through the accumulation of mutations, translation efficiency and replication speed are markedly suppressed [25,26]. Similarly, bovine viral diarrhea virus (BVDV) strains that establish persistent infection evolve at a slower rate than acutely infectious strains. The acutely infectious strain of BVDV is a highly virulent strain that replicates more rapidly, has a higher viral load, and is more virulent than persistently infectious strains [27–31]. Therefore, the phenotypic shift in RNA viruses defined in this study has been confirmed in epidemiologic studies. As with many evolutionary biological models, our general model can be applied across many taxa to interpret effects on the diversity of shape of viral load. It should be noted that some existing studies have already addressed within- and

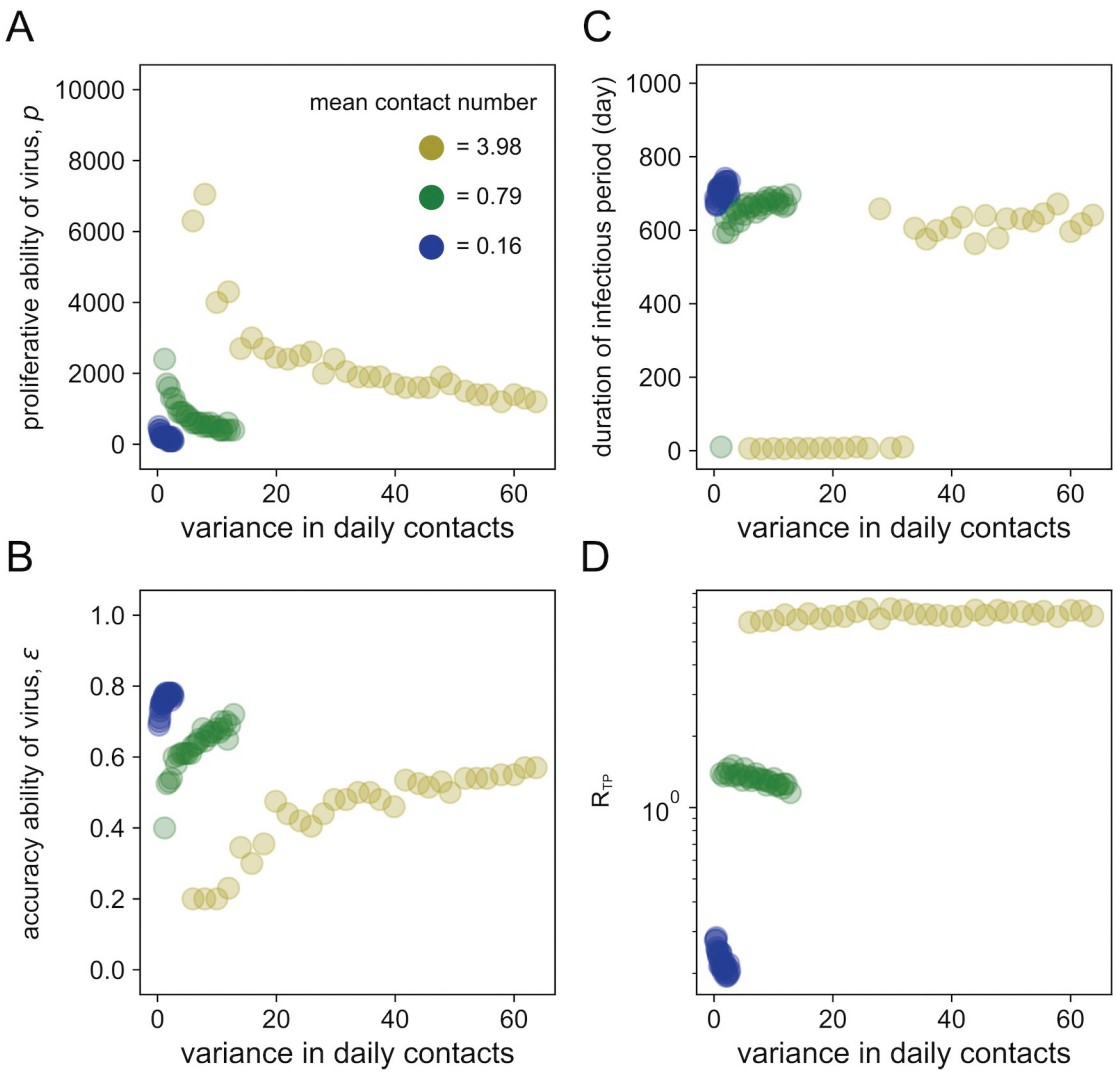

**Fig 4. Virus evolution in various scenarios of variance in daily contacts under three different mean contact numbers.** Each color represents the mean contact numbers of 3.98, 0.79, 0.16, respectively, corresponding to the colored dots in **Fig 3** in the main text. We used the following parameter sets: the scale parameter $\theta$ ranges from 0.5, 1.0, 1.5, . . ., 14.5, 15.0 (30 points in total), and the shape parameter $k$ ranges to keep the specific mean contact number. We conducted a total of 500 independent iterations for each parameter set of $k$ and $\theta$, and showed the mean values. For different variance in daily contact numbers shown in the horizontal axis, the optimal parameter combination of $p$ and $\varepsilon$ that maximizes $R_{TP}$ is plotted in (**A**) and (**B**), separately. The duration of the infectious period and $R_{TP}$ based on the parameters determined above are plotted in (**C**) and (**D**), respectively.

between-host(s) dynamics. There are, however, several differences between our model and previous ones.

First, although we explicitly defined the duration of infectious period as a function of viral load (Eqs (5–6)), some previous pieces of research considered the relationship between viral load and viral duration of infection under the invasion-persistence trade-off (e.g., [19,32]; also see [33] for immuno-epidemiologic multi-scale models that mechanistically capture the relationship). Those previous studies discussed a continuum of evolutionarily stable pathogen strategies. Pathogens evolve to greater acuteness to maximize between-host transmission in large well-mixed populations at one end, and selection favors less acute pathogens, which achieve enhanced between-population transmission when the host population is broken into

many small populations at the other end [20,34]. This study, without considering population structure of hosts, employs modelling of continuous evolution and phenotypic shifts of pathogens depending on mean and variance of contact rates, demonstrating the flexibility of adaptive evolution of pathogens that is observed in epidemiology in practice.

Second, research considering contact histories as an arena of virus evolution did not assume stochasticity [35] or extreme rare events [32]. There is a growing list of literature on the evolution of acuteness, especially shedding light on the importance of critical community size, which is approximately the population size above which pathogen extinction is not observed, implying an unbroken within-population chain of transmission (e.g., [36]). However, the previous research rarely considered variation in population size as a time series pattern. We adopted a negative binomial distribution to generate various contact histories, which corresponds to a broad range of host species' behavior. Using the current multi-level model, we especially focused on the condition where persistent viruses evolve. This study provides a detailed trace of how RNA viruses adapt to variable environments through the rare but significant events of beneficial mutations, which alter the phenotype and are inherited to the next generation of progeny viruses in a stochastic manner. As these mutations subsequently spread into the host population, they become fixed in the viral population, thereby shedding valuable insight into the complex stochastic process of viral evolution.

The acute and persistent infection phenotypes are classified in terms of the duration of the infectious period and the peak viral load, which is determined by the combination of the accuracy of genome RNA replication, $\varepsilon$, and its production rate, $p$. The acute infection phenotype (with a short infectious period and high peak viral load) is achieved in the parameter range where the production rate of virus particles is high and the replication accuracy is low. Although our model does not explicitly include the mutation process by which new viral phenotypes are generated (cf. [32]), this corresponds to a scenario with a low accuracy of replication in the current model, which is the case for influenza viruses with around 90% of noninfectious activities [37,38]. On the contrary, for the persistent infection phenotype, a long infectious period with a low peak viral load is obtained when there is high fidelity of viral replication and a low rate of genome replication. The properties characterized in this parameter region correspond to pathogens such as bornaviruses [39,40].

Our main finding is that the acute infection phenotype is optimal under conditions of a contact history involving frequent contacts between hosts (**Figs 2AB** and **3**). In this environment, the virus does not need to stay in the infected host for a long time, because the focal infected individual can quickly obtain another opportunity to infect (called a "leaving-home" strategy in [41]). Therefore, an effective strategy is to sustain a high peak viral load and adopt an increased probability of infection even for a short period of time (**Fig D in S1 Text**). On the other hand, in an environment with a small mean and a large variance in contact history, since there is a possibility that no contact will occur at all in a short period of infection (**Figs 2CD** and **3**), the virus must stay in the infected host for a long time (called the "stay-at-home" strategy in [41]). Hence, $R_{TP}$ increases by maintaining an infectious period with a long duration while suppressing the peak viral load (**Fig D in S1 Text**). This corresponds to the evolution of the persistent infection phenotype. Thus, the contact history of infected hosts determines the direction of viral evolution balanced between the duration of infectious period and peak viral load. That is, persistent-acute infection phenotype switching is induced depending on the environment.

In this study, one of the parameters that influenced the infection phenotype was the replication accuracy $\varepsilon$. The RNA-dependent RNA polymerase encoded by RNA viruses lacks proofreading activity, unlike DNA viruses. Owing to this low fidelity of viral polymerase, the genome sequence exists as diversified "quasispecies" in infected individuals [42–44]. These

intra-host quasispecies are involved in viral fitness *in vivo* and affect the virulence and transmission ability of a virus [4,42,45]. Interestingly, low-fidelity mutants of coxsackievirus B3, which primarily infects the heart muscle, have lost the ability to establish persistent infection *in vivo* [14]. This result on the relationship between the fidelity of viral polymerase and persistent infection *in vivo* are consistent with our study and support the model that replication accuracy $\varepsilon$ determines the establishment of the persistent infection phenotype.

Note that we can also easily extend our interpretation of $\varepsilon$ by including multiple factors such as the accuracy of virion assembly, the rate of virus particle aggregation, and others, in addition to the main factor of the accuracy of replication (see Eqs (1–4)). From the assumption that the virus is optimized to increase its transmission potential, the genetic algorithm heuristically searches for the optimal $R_{TP}$ as a population-based method with a group of solutions made up of combinations of $p$ and $\varepsilon$. In other words, $p\varepsilon I(t)$ in our model represents "infectious virus production", meaning the parameter $\varepsilon$ can include the multiple above-mentioned factors.

From an evolutionary perspective, organisms, including viruses, adapt to the environments they experienced in the past, and their traits are optimized accordingly [46]. Our exhaustive simulations with the multi-level model suggest that the acute infection phenotype is more likely to be optimal when individuals live in high densities or when there is a highly connected network of hosts (as is often the case for humans and other social animals). On the other hand, the persistent infection phenotype may be optimal in organisms that do not form groups and where the environmental variance for the virus is largely dependent on their life history; for example, if the contact rate increases only during the breeding season. The unpredictability of the presence of any individual, which is referred to "biotic drift", can influence evolutionary consequences [47]. The current model can be extended to analyze the seasonal (periodic) forcing in contact rates without loss of generality by assuming that the average value in the contact history is a periodic function.

Our simulations also show that the evolutionary endpoints of $R_{TP}$ are larger in the acute infection phenotype than in the persistent infection phenotype (**Fig 3E**). For example, with low-density contact histories ranging from 0.01 to 0.57, $R_{TP}$ decreases below 1, meaning that viruses with the persistent infection phenotype are unlikely to be sustained from the perspective of population-level transmission, even though the virus phenotype adapts to the low-density contact histories. This implies that acute virus infection, such as with SARS-CoV-2, will easily prevail in human society. BDV, an RNA virus that can establish persistent infection without cytopathic effects for a long time, has been observed as an endemic disease in Germany but has not reached a large-scale epidemic [22,48,49]. On the other hand, even if viruses with the persistent infection phenotype are not transmitted among individuals in the context of low-density contact history, an increase in contact numbers may lead to an optimized virus with the acute infection phenotype, increasing transmission potential within a host population. In fact, many acute infectious viruses that infect humans have been proposed to have been optimized through continuous mass infection after the formation of highly dense human communities for cultural reasons, such as for agriculture [50–53].

Large-scale genome sequencing and phylogenetic analysis of SARS-CoV-2 during the epidemic further the conclusions of this study. Epidemiologic studies have shown that SARS-CoV-2 evolved from pre-alpha to alpha, beta, and delta variants via host-to-host transmission [54]. These variants of concern (VOCs) transmit more quickly in the host population compared to background, suggesting that they may have shorter intrinsic generation times [55]. Moreover, the molecular clock modeling using the GISAID database and Bayesian analysis suggested that the substitution rate of SARS-CoV-2 is increasing with the emergence of VOCs relative to the background [56]. In the above, it can be inferred that the continuous chain of

infection within the host population facilitates viral evolution and contributes to the establishment of further acutely infectious viral phenotypes.

In terms of evolutionary biology, maximizing "fitness" like $R_{TP}$ is used as a metric for evaluating the direction of evolution. For the sake of simplicity, in the current study, we considered only the evolution of the virus at an epidemiologic equilibrium. We ignored the evolution of the host for simplicity, although we expect this to be reasonable because the generation time of a virus is much shorter than that of the host organism. However, to capture a long-term non-equilibrium evolution, it may be necessary to consider coevolutionary dynamics between hosts and viruses. A recent study [21] showed that pathogens often showing antigenic escape from host immune defenses evolve through non-equilibrium dynamics, indicating that $R_{TP}$ is no longer the measure to understand a viral evolutionarily stable strategy. Evolutionary game theory might be able to predict an evolutionarily stable strategy under the stochastic environmental scenario presented in this paper [57,58].

In addition, our approach has several limitations. For example, we do not explicitly include a host immune response or antiviral treatment, both of which may affect the duration of the infectious period and peak viral load. However, we do not expect this to change our qualitative conclusions, as these effects may be mimicked by changing the values of the parameters of our model. For example, the effect of cellular and humoral immune responses are implicitly included in the death rate of the infected cell (i.e., $\delta$ in Eq (3)) and the virus infection rate (i.e., $\beta$ in Eqs (1–2)), respectively. In our mathematical model, the viral propagation is limited by target cell availability rather than these immune responses. In fact, it seems that the target cell limitation is particularly important for infectious diseases in which the host's immune response can reduce symptoms or negative health outcomes but does not effectively suppress viral replication.

Our mathematical model for population-level transmission, Eqs (7–8), does not account for "a transmission chain", that is, we focused only on the sum of secondary cases generated throughout the infectious period by a primary case, $R_{TP}$. If entire chains of transmission are considered, it may be necessary to consider stochastic effects that may lead to particular viral phenotypes failing to become established in host populations. This would happen particularly in the region of a low-density contact history (the mean contact number from 0.01 to 1) for which $R_{TP} \approx 1$ [59]. Future studies may explore this further by nesting a viral dynamics model within a stochastic population-scale transmission model (using, for example, a stochastic adaptation of the multi-scale epidemiologic modeling framework proposed by [60]). Furthermore, explicitly considering transmission mode, host heterogeneity and detailed life history may also impact the outcomes of viral evolution qualitatively.

Overall, we anticipate our study to be a key advance in modeling theory that associates the type of virus optimized to the host environment. We face emerging respiratory acute virus infections such as SARS-CoV-2, and host environments continue to change as transportation networks are developed and urban areas expanded. In particular, the frequency of contacts between people has increased yearly. As we showed here, such an environment favors acute viral infections. To prevent such emerging infectious diseases, it is crucial that infectious disease surveillance systems be strengthened, particularly in locations with high contact densities. In such a way, extracting the contact history of a host allows us to extract the predict and understand the evolution of future viruses and introduce interventions quickly to prevent the transmission of novel viruses and variants. This is of clear benefit for public health.

## Materials and methods

To understand the dependence of viral evolution on the "environment", we employed a multi-level population dynamics model (i.e., coupling a population-level virus transmission model

and an individual-level virus infection model) and evaluated how the proliferative ability and accuracy of replication of viruses optimized to increase the transmission potential, $R_{TP}$ (see **Fig 1A**). For the sake of simplicity, we considered an epidemiologic equilibrium. We assumed that environments are characterized by the number of contacts per day (i.e., contact numbers among populations), and that an infected individual has a probability of infection (per contact with a susceptible individual) that depends on viral load. Note that the "noninfectious" period of infected individuals has not been directly accounted for in the infection probability (see below). We assumed that a newly infected individual has the potential to spread the infection to others immediately after infection, although the probability of infection is very small during this period. Below, we describe the details of the model as separate sections.

### Virus infection dynamics

We employed a simple mathematical model for virus infection dynamics considering the accuracy of viral replications [61]:

$$\frac{dT(t)}{dt} = \lambda - \beta T(t)V_i(t) - \mu T(t), \tag{1}$$

$$\frac{dI(t)}{dt} = \beta T(t)V_i(t) - \delta I(t), \tag{2}$$

$$\frac{dV_i(t)}{dt} = p\varepsilon I(t) - cV_i(t), \tag{3}$$

$$\frac{dV_{\mathrm{non}}(t)}{dt} = p(1-\varepsilon)I(t) - cV_{\mathrm{non}}(t), \tag{4}$$

where the variables $T(t)$, $I(t)$, $V_i(t)$ and $V_{\mathrm{non}}(t)$ are the number of uninfected target cells, infected target cells, and the amount of infectious virus (i.e., infectious viral load) and noninfectious virus at time $t$ since infection, respectively. The parameters $\beta$, $\delta$, $p$, and $cc$ represent the rate constant for virus infection, the death rate of infected cells, the per-cell viral production rate (i.e., proliferative ability of virus), and the per capita clearance rate of the virus, respectively. In addition, we assumed a fraction $\varepsilon$ of the produced virus is infectious but $1-\varepsilon$ is noninfectious (the value of $\varepsilon$ represents the accuracy of replication). To mimic viral cytopathogenesis depending on the viral replication level, we defined $\delta = \delta_{\max}p/(p+p_{50})$, that is, an increasing Hill function of $p$. The parameters $\delta_{\max}$ and $p_{50}$ are the maximum value of $\delta$ and the viral production rate satisfying $\delta = \delta_{\max}/2$, respectively. We fixed the parameter values (except for $p$ and $\varepsilon$) and initial values to correspond to biologically reasonable values or ranges (see below and **Table A in S1 Text**). The sensitivity analysis on $\delta$ is provided in **Fig G in S1 Text**.

### Duration of infectious period as a function of viral load

To describe the relation between the duration of infectious period and the viral load across a broad range of viruses (i.e., from acute phenotypes to persistent phenotypes) rather to explain properties of a specific virus, we assumed the duration of infectious period depends on the total viral load (i.e., the cumulative viral load) as observed for several pathogens [62,63]:

$$D(V_{\mathrm{total}}) = \frac{D_{\max}D_{50}^{D_K}}{V_{50}^{D_K} + V_{\mathrm{total}}^{D_K}}, \tag{5}$$

$$V_{\text{total}} = \int_0^{D_{\max}} V_i(t)dt, \qquad (6)$$

where $D_{\max}$ is the maximum duration of the infectious period, $V_{50}$ is the viral load at which the duration is half of its maximum, and $D_k$ is the steepness at which duration decreases with increasing viral load [63]. We also conducted a sensitivity analysis on the duration of the infectious period, using different function shapes instead of assuming Eq (5) (**Figs H, I, and J in S1 Text**). Note that other functions may be suitable to evaluate only a specific acute phenotypic virus, such as influenza virus or SARS-CoV-2, because the infected individual no longer sheds viruses after recovery or death.

## Probability of infection

To evaluate the daily transmissibility of an infected individual through the duration of infectious period, we assumed that for any given contact, each 10-fold increase in infectious viral load will lead to an $r$-fold increase in infectiousness [64]; thus, the probability of infection $\rho$ at time $t$ is described by:

$$\rho(t) = b \cdot r^{\log_{10}(V_i(t))}, \qquad (7)$$

where $b$ is the basilar probability of infection immediately after being infected with the virus.

## Number of encounters

To mimic a daily contact history, we assume the number of contacts on any day is drawn from a negative binomial distribution [65]. Since a negative binomial distribution is identical to a Poisson distribution where the mean parameter $\lambda$ follows a gamma distribution, the daily contact numbers $C(t)$ of a focal infectious individual at time $t$ are generated by sampling from the following distributions (see **Fig 1C** as examples):

$$\lambda \sim gamma(k, \theta),$$

$$C(t) \sim P_o(\lambda),$$

where $k$ and $\theta$ are the shape parameter and the scale parameter, respectively. We note that $k$ influences the skewness, and $\theta$ influences the variance of the distribution, respectively. The mean contact number through the history is the product of $k$ and $\theta$, that is, $k\theta$.

As a supplementary analysis, we also considered a different approach for generating the contact numbers $C(t)$, by considering a contact scenario on different networks: the Barabási–Albert (BA) scale-free network model and the Watts–Strogatz (WS) small-world network model (see the following sections: **Virus evolution based on network-generating contact Table B, Figs E, and F in S1 Text**). The BA model generates a network such that a graph of $n$ nodes is grown by attaching new nodes, each with $m$ edges, that are preferentially attached to existing nodes with high degree [66]. The WS model generates a network such that starting from a ring over $n$ nodes, which is connected with its $q$ nearest neighbors, shortcuts are created by replacing some edges with probability $p$ [67]. Contact numbers on a generated graph are calculated by counting the number of nodes that are connected with the one randomly chosen node on the network at each time step.

### Transmission potential and fitness landscape

We calculated the total number of secondary cases generated throughout the infectious period of a primary case ($R_{TP}$). On each day, the number of secondary cases is considered as the multiplication of the number of encounters (i.e., the contact number) and the probability of infection. Thus, $R_{TP}$ is calculated as below:

$$R_{TP} = \sum_{t=0}^{D(V_{\text{total}})} (C(t)\rho(t)), \tag{8}$$

where $D(V_{\text{total}})$ is the duration of infectious period of an infected individual defined by Eq (5). Once fixing the daily contact number, $R_{TP}$ is dependent on the parameters appearing in the mathematical model for the virus infection dynamics, Eqs (1–4). We hereafter focus on the dependence of the rate of proliferation and accuracy in the viral life cycle, $p$ and $\varepsilon$, respectively. We calculate $R_{TP}$ in various patterns of ($p$, $\varepsilon$), and thus the fitness landscape of $R_{TP}$ is constructed as a function of $p$ and $\varepsilon$.

### Evolution of virus to increase the transmission potential

One simple assumption is that the viral population will eventually be dominated by the virus with the largest $R_{TP}$. From this perspective, a genetic algorithm [68], a population-based method of evolutionary computation, is implemented to search for the optimal point in the fitness landscape (i.e., maximizing $R_{TP}$) regarding the combination of $p$ and $\varepsilon$. The genetic algorithm heuristically searches for the optimal $R_{TP}$ with a group of solutions made up of combinations of $p$ and $\varepsilon$ (see **Alg.1** and **Table B in S1 Text** for details). Initially, solutions are randomly placed around the fitness landscape (Initialize) and are assigned $R_{TP}$ as fitness (Evaluate). Over successive iterations, the combinations of $p$ and $\varepsilon$ with higher $R_{TP}$ are selected to build the next solutions (selectSolutions) in a fitness proportionate manner where the solution with higher $R_{TP}$ has a greater chance to be chosen for the next iteration. Note that the two highest-performing solutions are always chosen for the next iteration (Elitism). To find better combinations of $p$ and $\varepsilon$, solutions are recombined (crossover), and each parameter is slightly altered by adding a random variable drawn from a Uniform distribution with a pre-determined range (Mutate). Finally, the algorithm searches for a better solution with the expectation of finally converging to the combination of $p$ and $\varepsilon$ with the highest $R_{TP}$. These iterations can be considered as the virus "evolving" its properties of $p$ and $\varepsilon$ toward an optimal solution so that the virus can increase its own fitness in the evolutionary history.

### Virus evolution based on network-generating contact history

We investigate how the properties of a virus evolve based on contact history generated from the following well-established network models: Barabási-Albert scale-free network model (BA model) and Watts-Strogatz small world network model (WS model). The contact number of a contact history is the number of nodes connected to a node chosen at random from all nodes in the network (see **Table C in S1 Text** for the parameters used). As shown by 50 independent iterations, a virus under the contact history generated by both the BA model and the WS model evolves high virus production but low accuracy (**Figs EC and FC in S1 Text**), which corresponds to the characteristics of the acute infection phenotype.

```
Algorithm 1: Genetic Algorithm
Bb
Input: P(t = i) = {s₁, s₂, ..., sⱼ}, population of solutions in iteration
i
```

```
t ← 0;
Initialize (P(t = 0)) with random p and ε;
Evaluate (P(t = 0));
while not termination do
 P(t)_e ← Elitism (P(t));
 P(t)_p ← selectSolutions (P(t));
 P(t)_c ← crossover (P(t)_e +P(t)_p);
 Mutate (P(t)_c);
 Evaluate (P(t)_c);
 P(t+1) ← P(t)_c;
 t ← t + 1;
end while
```

## Supporting information

**S1 Text. Supplementary tables and figures.** Table A. Parameters used for virus infection dynamics in the main text. Table B. Parameters used in GA Algorithm. Table C. Parameters used for the network models. Fig D. Relation between duration of infectious period and peak viral load. Fig E. Evolution of virus on a contact history generated from the Barabási-Albert scale free network model. Fig F. Evolution of virus on a contact history generated from the Watts-Strogatz small world network model. Fig G. Virus evolution in various scenarios of contact history. Fig H. Different function types in the duration of infectious period assumed to depend on the cumulative viral load. Fig I. Virus evolution in various scenarios of contact history where a monotonically decreasing linear relationship is assumed between the duration of the infectious period and the cumulative viral load. Fig J. Virus evolution in various scenarios of contact history where a monotonically decreasing exponential relationship is assumed between the duration of the infectious period and the cumulative viral load
(DOCX)

## Author Contributions

**Conceptualization:** Ryo Komorizono, William S. Hart, Robin N. Thompson, Akiko Makino, Keizo Tomonaga, Shingo Iwami, Ryo Yamaguchi.

**Formal analysis:** Junya Sunagawa, Shingo Iwami, Ryo Yamaguchi.

**Funding acquisition:** Keizo Tomonaga, Shingo Iwami, Ryo Yamaguchi.

**Investigation:** Junya Sunagawa.

**Methodology:** Shingo Iwami, Ryo Yamaguchi.

**Software:** Junya Sunagawa.

**Supervision:** Shingo Iwami, Ryo Yamaguchi.

**Validation:** Junya Sunagawa.

**Visualization:** Junya Sunagawa, Shingo Iwami, Ryo Yamaguchi.

**Writing – original draft:** Junya Sunagawa, Ryo Komorizono, Hyeongki Park, William S. Hart, Robin N. Thompson, Akiko Makino, Keizo Tomonaga, Shingo Iwami, Ryo Yamaguchi.

**Writing – review & editing:** Junya Sunagawa, Ryo Komorizono, Hyeongki Park, William S. Hart, Robin N. Thompson, Akiko Makino, Keizo Tomonaga, Shingo Iwami, Ryo Yamaguchi.

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
