## [Decision Letter · Decision Letter 0]

19 Dec 2022

Dear Professor Iwami,

Thank you very much for submitting your manuscript "Contact-number-driven virus evolution: a multi-level modeling framework for the evolution of acute or persistent RNA virus infection" for consideration at PLOS Computational Biology.

As with all papers reviewed by the journal, your manuscript was reviewed by members of the editorial board and by several independent reviewers. In light of the reviews (below this email), we would like to invite the resubmission of a significantly-revised version that takes into account the reviewers' comments.

We cannot make any decision about publication until we have seen the revised manuscript and your response to the reviewers' comments. Your revised manuscript is also likely to be sent to reviewers for further evaluation.

Sincerely,

Joel O. Wertheim

Academic Editor

PLOS Computational Biology

Natalia Komarova

Section Editor

PLOS Computational Biology

Reviewer's Responses to Questions

**Comments to the Authors:**

Reviewer #1: The authors present a study examining the evolution of chronicity (the ability to cause chronic versus acute infections) in response to varying host contact patterns. Focusing on the case study of viruses, the authors construct a model linking within-host dynamics to variable contact rate patterns, where new hosts are contacted at a rate that either has a high mean and low variance or a low mean and large variance (subsequently extended to a range of means and variances). They then use simulations to determine the viral proliferation rate and accuracy that maximize R0, the number of hosts infected over the lifespan of the primary infection. The results suggest a tradeoff between proliferation rates and replication accuracy, where the optimal combination depends on contact rate patterns. When hosts contact each other at a rate with a high mean, that selects for fast proliferating viruses with low accuracy, while low rates of contact select for slow proliferating viruses with high accuracy. These results suggest that host contact patterns can exert a strong influence over traits expressed within the host, including proliferation rates.

The question of what drives the evolution of chronicity is a very important one, and the model used by the authors is simple in the best way. There are no extraneous details in the model structure (like the inclusion of immunity) that would make it difficult to understand the results. The figures are visually appealing and easy to understand, and I appreciate that the authors repeated their analysis for a large range of mean/variance relationships. While I admire the clarity of the approach, there are a few areas where the MS would benefit from revision:

- The authors could do much more to place this study in the context of previous work in this area. Past work has examined the evolution of replication rates while allowing infection duration to depend on viral load (King et al. 2009 Am Nat), which may necessitate rewording lines 200-202. That study and subsequent ones (Shrestha et al. 2013 Theoret Ecol) suggest important limitations to the idea that natural selection will always maximize R0, and those limitations would be worth discussing further in this MS (such as lines 104-106). There’s also a wealth of literature on the evolution of acuteness especially in the context of measles that would be good to discuss here (e.g., Gunning & Wearing 2013 Ecol Lett). Referencing this body of work—and making the unique contributions of this study very clear—would greatly enhance its impact.

- I realize that putting the methods after the results is part of this journal’s format, but more methods need to be outlined at the beginning of the results to make sense of what follows. I’d suggest adding a panel to Fig. 1 to illustrate the within-host model structure and make it obvious visually that the rate of transmission varies with viral load. One idea would be to arrow size constant within ta (and within tb and tc) but varying arrow size with viral load to emphasize that both transmission rates per contact and contact rates are changing over the course of infection, but only transmission rates per contact depend on viral load. Some might misinterpret Fig. 1A as showing that host contact rates vary with viral load (not an unreasonable assumption, but not what this model is doing). It would also be helpful to give R0 for this example (R0=4, if I’m reading the figure example correctly) so that readers can be sure they are following along.

- It is not clear to me why proliferation rates tradeoff with accuracy. The death rate of infected cells increases (but saturates) with increasing p, a cost of replicating faster. But why don’t viruses always evolve to maximize accuracy? My best guess is that fast proliferation, when combined with high accuracy, depletes target cell abundance and causes disfavorable fluctuations in viral load (i.e., causing low transmission rates that reduce the number of new hosts infected). If that guess is correct, it would be good to make that clearer, since an implicit tradeoff that arises from the biologically reasonable assumptions is (in my opinion) much more interesting than a tradeoff that was explicitly assumed at the outset.

- It would also be good to discuss a key counterexample, why HIV has been able to persist as a chronic infection despite the results in this MS suggesting that it should have evolved to be a faster-proliferating virus. My sense is that HIV violates a key assumption in the present model, that inaccurate replication does not impact infection duration. Mutations within the host could instead lead to variants that are better able to evade the immune system, despite their reduced ability to transmit (Lythgoe et al. 2017 Trends Microbiol). Omitting mention of this prominent counterexample might come across to readers as cherry-picking only case studies that agree with the model results.

Minor points:

Lines 255-256: Seasonal forcing in contact rates aren’t explicitly examined in this model, and it’s not clear how the results could be extrapolated to this case. Clarify or omit?

Lines 303-308: This phrasing undercuts the value of this study without adding useful nuance for readers. I agree with the authors that including immunity wouldn’t change the conclusions, but leading the reader through that logic would serve the authors better here. Could the authors say more about their assumption that viral proliferation is limited by target cell availability (bottom-up control) rather than immunity (top-down control)? It strikes me that target cell availability will be a problem for viruses whether or not hosts mount a useful immune response. Likewise, if immunity reduces symptoms/negative health outcomes but does not strongly impede viral replication (as in COVID-19), then target cell limitation takes on even greater importance.

Lines 323-326: It seems strange to discuss surveillance systems when the model has more relevance to contact rate interventions like social distancing.

Supplemental text S1 and S2: I'm not sure why this text is relegated to the supplement. Might be worth including in the main text?

Reviewer #2: Comments:

1. Throughout the paper, persistent viruses are characterized by low viral replication rate and decreasing evolutionary rate, which is not true in general. In particular in the introduction, they state "persistent infectious virus can coexist with the host by reducing the amount of progeny virus released from the infected cells, slowing down the replication rate, decreasing evolutionary rate, and evading the host's immune system''. Viruses such as HCV, referred to as an example, and HIV actually have high viral production and evolutionary rate, using this to evade the immune response and be persistent virus, but also include latent reservoir which might be a very nuanced case to characterize as persistent low replication. Further discussion of this strategy for some persistent viruses and how it reconciles with their framework and results would be helpful. While the authors do acknowledge in the Discussion that antigenic escape, not considered in this study, may change the methods and results, it seems that they are discussing coevolution between hosts rather than viral evolution within host against immune response.

2. There is a lack of clarity in how the parameter \\epsilon is being utilized in the evolutionary modeling. First, the authors give nebulous definitions of \\epsilon, saying one line 120, it "includes multiple-factors such as the accuracy of virion assembly, the rate of virus particle aggregation and others, in addition to the accuracy of replication which is the main factor. However the parameter is seemingly a simple probability of viral progeny being "infection capable" in Eq. 3-4. It also seems that \\epsilon is a completely independent parameter, as there is no explicit mention of a tradeoff between \\epsilon and viral production rate p. Nevertheless, the Results seem to suggest that the authors are assuming some kind of tradeoff between \\epsilon and viral production rate p (the contour lines in p vs \\epsilon fitness graphs shown in Fig. 1B, 2). They do have a tradeoff formula between parameters p and infected cell death rate \\delta, but \\epsilon is not in this formula. It would make sense to have all these tradeoffs, in particular without it, one would expect viruses to always optimize their fitness by maximizing accuracy at \\epsilon=1. I suppose that the infection duration being a decreasing function of total viral load (Eq. 5) could be the tradeoff, but the authors need to clarify and discuss. Also, this assumption in Eq. 5 needs further discussion as it seems critical for the results.

3. The mutation rate or precise formulation used in genetic algorithm and evolutionary computations are not given. Also one might think of this being tied to accuracy of replication, which is one of the terms describing \\epsilon above.

4. In Discussion, authors state "none of previous researches assumed the relationship between viral load and viral duration of infection''. However, it should be noted that several works have considered immuno-epidemiological multi-scale models for evolution of pathogen with immune response in within-host model, which consider various forms of recovery rates that mechanisticly capture the relationship between viral load and duration of infection; e.g. "Gulbudak H, Cannataro VL, Tuncer N, Martcheva M (2017) Vector-borne pathogen and host evolution in a structured immuno-epidemiological system. Bull Math Biol 79(2):325–355".

**Have the authors made all data and (if applicable) computational code underlying the findings in their manuscript fully available?**

Reviewer #1: Yes

Reviewer #2: None

PLOS authors have the option to publish the peer review history of their article (what does this mean?). If published, this will include your full peer review and any attached files.

Reviewer #1: No

Reviewer #2: No
---

## [Decision Letter · Decision Letter 1]

27 Mar 2023

Dear Professor Iwami,

Thank you very much for submitting your manuscript "Contact-number-driven virus evolution: a multi-level modeling framework for the evolution of acute or persistent RNA virus infection" for consideration at PLOS Computational Biology. As with all papers reviewed by the journal, your manuscript was reviewed by members of the editorial board and by several independent reviewers. The reviewers appreciated the attention to an important topic. Based on the reviews, we are likely to accept this manuscript for publication, providing that you modify the manuscript according to the review recommendations.

The manuscript is much improved. One review still had some concerns, primary around clarity and narrative. Please address their comments in full.

Sincerely,

Joel O. Wertheim

Academic Editor

PLOS Computational Biology

Natalia Komarova

Section Editor

PLOS Computational Biology

The manuscript is much improved. One review still had some concerns, primary around clarity and narrative. Please address their comments in full.

Reviewer's Responses to Questions

**Comments to the Authors:**

Reviewer #1: The authors have addressed my main concerns, but there are still some places where the narrative could be better explained and justified:

Fig. S4 seems to have some other patterns worth mentioning in the main text, including that large variance in daily contacts can select for greater accuracy/lower proliferative capacity (light green dots). The authors might consider moving this figure to the main text, especially since the statement they make in lines 192-195 requires the support of Fig. S4.

The paragraph beginning on line 210 seems like a strange addition to the results, but would work well in the discussion, perhaps as the first paragraph. In general, the discussion could use a bit more polishing, and the current first paragraph lacks citations and does not add much insight. The paragraphs giving context to previous work are helpful, but it would be even better to end each of those paragraphs by linking the present model results. For example, in the paragraph beginning in line 239, how does examining the mean and variance of contact rates add insight to what has already been learned from comparing large well-mixed populations to small, isolated ones? I’m not sure line 252-253 is true, since lot of models consider seasonal migration patterns of human movement. It would also be more useful to focus on what insight is gained by incorporating stochasticity and extreme rare events, else the paragraph beginning in line 248 reads as unconstructive criticism. One of the strengths of this MS is jointly considering accuracy and replication rates, and that strength is undersold in the two paragraphs referencing past work.

In the response to Reviewer 2, the authors mention the key assumption that the duration of the infectious period decreases with cumulative viral load. This assumption is not mentioned until the discussion, and it’s necessary to understand the results. The explanation in lines 143-144 does not make this clear, and I suggest adding the text in this response to the first paragraph of the results, because it would greatly clarify this point of confusion:

“…we did not make any assumptions about a trade-off between the accuracy ability of virus () and the proliferative ability of virus (), while it is

assumed that the death rate of virus-producing cells () depends on the proliferative ability of virus () due to viral cytopathogenesis. In contrast, we assumed that the duration of the infectious period decreases with the cumulative viral load, which is a crucial assumption in our study. This is because epidemiological studies have shown that patients with severe disease and a short duration of infection tend to have a high viral load, while those with relatively mild disease tend to have a low viral load [24-26]. Under these biologically reasonable assumptions, the trade-off in the duration of infectious period, as shown in Fig. 1B, arises from our individual-level virus infection model, as described in Eqs.(1- 4).”

Minor comments:

Line 128-130: Would be helpful to break this sentence down and go through the logic more gently.

Line 140: Suggest omitting “mainly reflecting into” for clarity

Line 135: Would be good to start with something like what is currently in the methods (lines 378-389), so that readers have some idea where the results are coming from.

Figure 1 caption: The new figure is very helpful but it would be even clearer if the caption referenced mathematical terms, for example making line 747-8 read “The probability of infection (rho) of susceptible individuals from an infected hosts depends on their viral load (Vi(t)).” Likewise, line 750 could read “…the contact numbers (c(t)) are assumed to follow…”

The explanation from lines 192-208 was a little hard to follow because the text refers to unlogged contact rates and the figures show logged contact rates. Maybe keep the log scale in Fig. 3 but replace labels with unlogged values?

Lines 345-347: suggest omitting these sentences—no model (or experiment) will ever fully reflect the detailed physiologic processes occurring within the host in a natural setting.

Line 356—Start a new paragraph with “Our mathematical model for population transmission…” [Omit the “Another modeling limitation”—no need to apologize for not considering everything in a single study.]

Lines 371-375—Split up this sentence for clarity.

Reviewer #2: The reviewers have answered the comments and edited accordingly.

**Have the authors made all data and (if applicable) computational code underlying the findings in their manuscript fully available?**

Reviewer #1: Yes

Reviewer #2: None

PLOS authors have the option to publish the peer review history of their article (what does this mean?). If published, this will include your full peer review and any attached files.

Reviewer #1: No

Reviewer #2: No

Figure Files:

Data Requirements:

Reproducibility:

References:

---

## [Editor Report · Decision Letter 2]

10 May 2023

Dear Professor Iwami,

We are pleased to inform you that your manuscript 'Contact-number-driven virus evolution: a multi-level modeling framework for the evolution of acute or persistent RNA virus infection' has been provisionally accepted for publication in PLOS Computational Biology.

Best regards,

Joel O. Wertheim

Academic Editor

PLOS Computational Biology

Natalia Komarova

Section Editor

PLOS Computational Biology

---

## [Editor Report · Acceptance letter]

22 May 2023

PCOMPBIOL-D-22-01431R2 

Contact-number-driven virus evolution: a multi-level modeling framework for the evolution of acute or persistent RNA virus infection

Dear Dr Iwami,

I am pleased to inform you that your manuscript has been formally accepted for publication in PLOS Computational Biology. Your manuscript is now with our production department and you will be notified of the publication date in due course.

With kind regards,

Zsofia Freund
